# Image Segmentation using Transfer Learning with DeepLabv3 to Facilitate Photogrammetric Limb Scanning

## Abstract

In this paper, we explore the use of deep learning (DL) in conjunction with photogrammetry for scanning amputated limbs. Combining these two technologies can expand the scope of prosthetic telemedicine by facilitating low-cost limb scanning using cell phones. Previous research identified image segmentation as one of the main limitations of using photogrammetry for limb scanning. Based on those limitations, this work sought to answer two main research questions: (1) Can a neural network be trained to identify and segment an amputated limb automatically? (2) Will segmenting 2D limb images using neural networks impact the accuracy of 3D models generated via photogrammetry? To answer the first question, transfer learning was applied to a neural network with the DeepLabv3 architecture. After training, the model was able to successfully identify and segment limb images with an IoU of 79.9%. To answer the second question, the fine-tuned DL model was applied to a dataset of 22 scans comprising 6312 limb images, then 3D models were rendered utilizing Agisoft Metashape. The Mean Absolute Error (MAE) of models rendered from images segmented with DL was 0.57 mm ± 0.63 mm when compared to models rendered from ground truth images. These results are important because segmentation with DL makes photogrammetry for limb scanning feasible on a large clinical scale. Future work should focus on generalizing the segmentation model for different types of amputations and imaging conditions.

## 1    Introduction

Rehabilitative care for persons with limb loss is rapidly evolving due to advances in digital healthcare technologies. Novel digital workflows are empowering clinicians with tools for visualizing patient anatomy and physiology, designing custom fitting prostheses via computer aided design (CAD), building assistive devices with computer aided manufacturing (CAM), and tracking patient response in environments such as virtual reality (VR) Cabrera et al. (2021). Medical imaging technologies are fundamental to every digital workflow because they inform clinicians of limb geometry, surface and/or sub-surface features, plus pathology of amputated limbs Paxton et al. (2022).

Systematic reviews by Cabrera et al. (2021) and Paxton et al. (2022) identified photogrammetry as a promising technology for capturing patient surface anatomy. The main advantage of photogrammetric scanning is that models can be rendered using photographs captured via smartphones Cabrera et al. (2020); Barbero-García et al. (2018); De Vivo Nicoloso et al. (2021); R. B. Taqriban et al. (2019); Ismail et al. (2020); Barbero-García et al. (2020; 2021). Scanning with smartphones is significantly cheaper than other medical imaging modalities Cabrera et al. (2021); Paxton et al. (2022) and results in reliable and robust surface accuracy on par with existing clinical gold standard technologies Nightingale et al. (2020; 2021). Unfortunately, photogrammetry workflows often require extensive image segmentation, at the expense of human operator time and effort, in order to render 3D models Cabrera et al. (2021).

Segmentation is an important problem in medical imaging and involves separating regions of interest (ROIs) from the rest of an acquired image. Convolutional neural networks (CNNs) are regarded as the dominant state-of-the-art approach for medical image segmentation in applications requir-

ing high-accuracy Kumar et al. (2020); Wang et al. (2022). Deep convolutional neural networks (DCNNs), such as DeepLabv3, are able achieve high IoU performance when classifying pixels and outperform other CNN architectures Chen et al. (2017b). Using transfer learning, it is possible to fine-tune pre-trained deep neural networks with instances from the target domain Zhuang et al. (2020). Transfer learning is crucial to medical imaging because in many cases it is not possible to collect sufficient training data Kumar et al. (2020); Wang et al. (2022); Zhuang et al. (2020).

We hypothesize that automating image segmentation via DeepLabv3 then rendering the segmented images using photogrammetry could create an efficient processing pipeline for scanning amputated limbs with smartphones. Automatic segmentation of limb photographs would allow for more photographs to be taken during the scanning procedure thus increasing the sampling density. With these additional photographs, it would be possible to improve the coverage and accuracy of 3D models. Finally, segmentation could help correct for motion of the limb during the scanning procedure. These potential benefits would allow photogrammetric limb scanning to be used on a larger scale to reach more patients in the clinic and remotely via telemedicine.

## 2 BACKGROUND

### 2.1 PHOTOGRAMMETRY FOR MEDICAL IMAGING

In its simplest form, photogrammetry is the science of measuring size and shape of objects using images Fryer (1996). In this context, photogrammetry has been used extensively since the 1950's for medical imaging and measurementNewton & Mitchell (1996). The development of digital cameras and the accompanying transition to digital photogrammetry has led to technologies for the reconstruction of 3D models using photogrammetric algorithms Linder (2016). Digital photogrammetry has been used successfully to reconstruct patient anatomy in many medical contexts such as cranial deformation scanning Barbero-García et al. (2017; 2020; 2021), facial scanning Ross et al. (2018); Nightingale et al. (2020; 2021), and amputated limb scanning R. B. Taqriban et al. (2019); Cabrera et al. (2020); Ismail et al. (2020); De Vivo Nicoloso et al. (2021).

Two values for accuracy are commonly reported for photogrammetric models: Root Mean Squared Error (RMSE) and Mean Absolute Error (MAE). RMSE values will always be greater than or equal to MAE values and are more susceptible to outliers, but are recommended when errors are unbiased and follow a normal distribution Chai & Draxler (2014). Close range photogrammetric approaches have been proven to have accuracy comparable to clinical gold standard technologies Ross et al. (2018).

Using an Artec Spider structured light scanner as a clinical "gold standard" reference, Nightingale et al. (2020) found that photogrammetric reconstructions of facial scans using 80 images captured had RMSE accuracy values of 1.3 mm ± 0.3 mm. In a similar study, Nightingale et al. (2021) achieved RMSE accuracy values of 1.5 mm ± 0.4 mm with 30 photographs on reconstructions of the external ear. Using spherical objects with known geometry as a reference, Barbero-García et al. (2018) was able to achieve an MAE accuracy of 0.3 mm ± 0.2 mm using 95 images with tie point aids. In later research, these authors were able to achieve similar accuracy while scanning infant skulls with MAE accuracy values of 0.5 mm ± 0.4 mm and 200 images Barbero-García et al. (2020).

While the accuracy of photogrammetry for anatomical scanning is very good, workflows involving photogrammetry require a great deal of human input often taking hours Barbero-García et al. (2017); Cabrera et al. (2021). For this reason, recent research has focused on various methods for automating photogrammetric workflows for anotomical scanning Barbero-García et al. (2020); Cabrera et al. (2020). Photogrammetric models are rendered following acquisition (not in real time) thus errors in the image acquisition stage may not become evident until after a patient is scanned and no longer present. Automated approaches have focused their attention on this acquisition stage to ensure completeness of the results Nocerino et al. (2017) with recent advances incorporating machine learning for landmark detection Barbero-García et al. (2021).

Still, automation of photogrammetric image processing (specifically image segmentation) remains a large problem Cabrera et al. (2021). Automating this image segmentation step could dramatically increase the speed of photogrammetric workflows for medical imaging, improving the clinical viability.

## 2.2 Image Segmentation with Deep Learning

Medical image segmentation plays an essential role in modern clinical practice by facilitating computer aided diagnoses and making patient anatomical structures clear Wang et al. (2022). Prior to recent developments in deep learning (DL), computer vision techniques such as k-nearest neighbors (kNN), decision trees (DT), support vector machines (SVM), and random forest (RF) models were utilized for segmentation and classification tasks Thanh Noi & Kappas (2017) Mahony et al. (2019). DL has superseded all of these approaches in medical imaging for several reasons, but most notably because the burden of feature engineering in DL shifts from humans to computers Shen et al. (2017).

CNNs are the most heavily researched DL architectures for medical image analysis J. Ker et al. (2018). CNN architectures are composed of several fundamental building blocks: Convolution Layers, Pooling Layers, Activation Functions, Fully Connected Layers, and Loss Functions Alzubaidi et al. (2021). Convolution Layers are the most significant portion of CNNs and comprise a collection of convolutional filters (referred to as kernels). Input images are convolved using these filters to create an output feature map. Pooling Layers sub-sample the feature maps effectively shrinking them to create smaller feature maps. They also enlarge the receptive fields of neurons in subsequent layers allowing for translation invariance of the architecture. Activation Functions introduce non-linearity into the neural network and map inputs to outputs by selectively choosing to fire neurons. Fully Connected Layers are typically located at the end of the CNN architecture, with inputs from the final convolution or pooling layer, and are used as the CNN classifier. Loss functions are typically utilized in the output layer to calculate the predicted error across training samples Alzubaidi et al. (2021); Goodfellow et al. (2016); Arif (2020).

The DeepLabv3 architecture expands on traditional CNNs by utilizing atrous convolution with spatial pyramid pooling Chen et al. (2017b). Atrous convolution utilizes a dilated convolution kernel that, rather than focusing on adjacent pixels, convolves pixels that are not immediate neighbors. Atrous convolution helps to resolve the issue of spatial resolution loss in feature maps from successive convolution and de-convolution steps by allowing for precise control of feature map density. Atrous Spatial Pyramid Pooling (ASPP) Chen et al. (2017a) utilizes four parallel atrous convolutions with different atrous rates to transform feature maps. This allows for accurate and efficient classification regions at arbitrary scales Chen et al. (2017b). These elements of the DeepLabv3 architecture result in better classification of pixels for semantic image segmentation, on par with other state-of-the-art models.

CNNs are prepared for medical imaging tasks via three major techniques: 1) Training CNNs from scratch using large datasets of labeled images. 2) Using "off-the-shelf" features without retraining the CNN to complement hand crafted features. 3) Pre-training the CNN on natural or medical images, then fine-tuning on target images (i.e. transfer learning) Shin et al. (2016). Transfer learning is commonly used to reduced the computational time and cost of training a neural network from scratch Mahony et al. (2019); Wang et al. (2022); Shen et al. (2017). In medical imaging tasks, transfer learning has been shown to be as good as training CNNs from scratch and better in many cases Tajbakhsh et al. (2016); Shin et al. (2016). Recent research has shown great success in using transfer learning with the DeepLabv3 architecture for medical imaging segmentation Roy et al. (2020).

## 3 Methods

### 3.1 Transfer Learning using DeepLabv3

We performed transfer learning on a pre-trained resnet-101 based DeepLabv3 model downloaded from the Pytorch Model Zoo Paszke et al. (2019). This model was pre-trained on the COCO train2017 data set comprising 200,000 images and 80 object categories Lin et al. (2015). Our fine-tuning dataset consisted of 806 manually labeled images of limbs with transtibial amputations. This data was not augmented. These photographs were taken in a variety of conditions (e.g. different angles, indoor, outdoor, bare skin, with liners etc.). Images were downsampled from 3072 x 4096 to 300 x 400. Following this, we randomly select 80% of the images as our training data set and reserve the remaining 20% for validation.

We used Google Colab for training the model with dynamically scaled computer resources. In terms of the hyperparameters in fine-tuning, we used the Adam algorithm Kingma & Ba (2017) for optimizing the parameters (e.g. weights) in the neural network with the default learning rate being set to $10^{-5}$. We employed the pixel-wise Mean Squared Error (MSE) as our loss function. We trained our fine-tuned neural network for 10 epochs using a batch size of 8 within each epoch.

## 3.2 Scanning Workflow

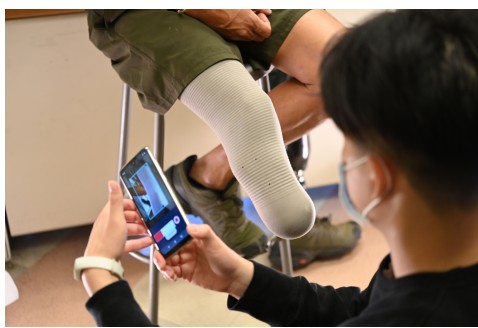

Figure 1: Scanning a Transtibial Amputee using the Lim(b)itless Cell Phone Application

We captured 24 limb scans to test the applicability of the neural network model for image segmentation. 12 scans were performed with a single transtibial amputee wearing a standard liner then 12 additional scans were performed with the transtibial amputee wearing a colored sock over their standard liner. Scans were performed utilizing the Lim(b)itless smartphone application Cabrera et al. (2020) on a Motorola Moto G7 (MSRP $299, Camera - 12 MP, f/1.8, 1/2.8") see Fig. 1.

The scanning procedure involved making two revolutions (one clockwise, one counterclockwise) around the outside of the limb while keeping the limb in frame of the application. Scans were taken at a constant radius of 30 cm from the limb, similar to Barbero-García et al. (2018), at a rotation rate of two revolutions per minute. This application automatically initiates a capture sequence at 10 degree intervals with each capture sequence including a burst of 3 photographs. For two full revolutions, this captures a hyper-redundant 216 images (minimum) per scan as recommended by Barbero-García et al. (2020). Actual scanning time averaged 52.4 s ± 6.1 s while the actual number of images captured averaged 287 ± 29, Table 1.

Table 1: Image acquisition and rendering pipeline statistics for smartphone scans (n = 22) used in this study.

| | |
|---|---|
| **Number of Images** | 287 ± 29 |
| **Scan Time (s)** | 52.4 ± 6.1 |
| **Segmentation Time (s)** | 890 ± 111 |
| **Render Time (s)** | 3430 ± 850 |
| **Scan Rate (images/s)** | 5.54 ± 0.74 |
| **Segmentation Rate (images/s)** | 0.32 ± 0.01 |
| **Render Rate (images/s)** | 0.088 ± 0.022 |

## 3.3 Rendering Procedure

All segmentation and rendering was performed on a desktop server running an Intel(R) Xeon(R) CPU E5-1607 v3 3.10 GHz, Nvidia Quadro k2200 4 GB DDR5, and 256 GB DDR4 RAM 1866 MHz. Of the 24 limb scans acquired in the previous step, two scans were manually segmented via Adobe Photoshop and set aside as ground truth references. The remaining 22 scans, totaling 6312 images, were fed into the fine-tuned neural network for segmentation. Segmentation time using the neural network averaged 890 s ± 111 s, Table 1.

3D limb models were rendered from these segmented images using Agisoft Metashape, Fig. 2. Agisoft is the most commonly used photogrammetric software in medical imaging being used by

Cabrera et al. (2020); Barbero-García et al. (2018; 2017); Nightingale et al. (2020; 2021). This step of the workflow was automated with no manual tie point alignment or point cloud segmentation. The resulting STL meshes were not smoothed prior to export, unlike Ross et al. (2018); Nightingale et al. (2020; 2021). This Laplacian smoothing step was foregone to evaluate the impact of segmentation method on model rendering.

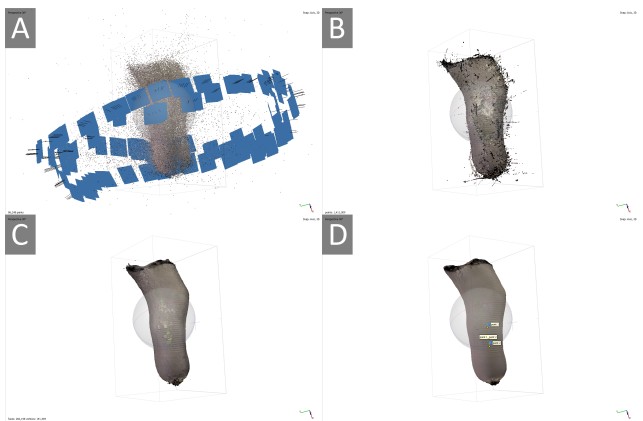

Figure 2: Procedure for rendering limbs in Agisoft Metashape: A) Photo alignment and tie point detection. B) Creation of a dense point cloud. C) Conversion of point cloud to STL mesh. D) UV texture mapping and model scaling.

Finally, STL meshes of the scans rendered from automatically segmented images were compared with scans rendered from ground truth images via the CloudCompare open source 3D point cloud and mesh processing software. All limbs were cropped to the same boundaries, and outliers (visualized as extraneous floating points) were filtered out consistent with the technique used by Ross et al. (2018). Meshes were compared point by point via the cloud-to-mesh (C2M) measurement tool, with ground truth models being set as a reference.

## 4 RESULTS AND DISCUSSION

### 4.1 SEGMENTATION VIA TRANSFER LEARNING

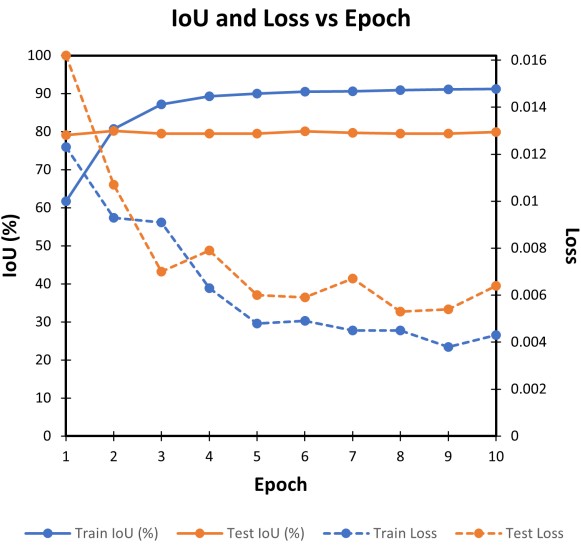

Figure 3: Evolution of IoU and loss function for training and testing dataset

Table 2: Training and testing IoU metrics and loss function evolution over the course of transfer learning.

| Epoch | Train IoU (%) | Test IoU (%) | Train Loss | Test Loss |
|---|---|---|---|---|
| 1 | 61.7 | 79.1 | 0.0123 | 0.0162 |
| 2 | 80.7 | 80.2 | 0.0093 | 0.0107 |
| 3 | 87.2 | 79.5 | 0.0091 | 0.0070 |
| 4 | 89.3 | 79.5 | 0.0063 | 0.0079 |
| 5 | 90.0 | 79.5 | 0.0048 | 0.0060 |
| 6 | 90.5 | 80.1 | 0.0049 | 0.0059 |
| 7 | 90.6 | 79.7 | 0.0045 | 0.0067 |
| 8 | 90.9 | 79.5 | 0.0045 | 0.0053 |
| 9 | 91.1 | 79.5 | 0.0038 | 0.0054 |
| **10** | **91.2** | **79.9** | **0.0043** | **0.0064** |

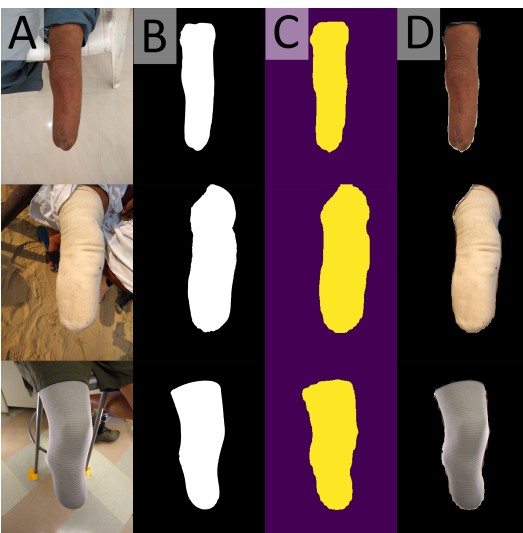

Figure 4: Limbs with transtibial amputations used in the transfer learning process. A) These limb photographs were taken in a variety of lighting conditions and with different liners. B) Manually segmented binary masks representing the ground truth. C) Masks generated by the fine-tuned neural network. D) Applying the masks from the fine-tuned neural network to the original images.

Fig. 3 and Table 2 show the evolution of the fine tuned neural network over the transfer learning process. After 10 epochs of training, the fine tuned neural network showed excellent performance in segmenting images of transtibial amputations, see Fig. 4.

Looking at the training IoU curve, it can be observed that the training IoU performance increases with model epochs. This is to be expected as the training loss curve simultaneously decreases, reaching a local minimum at epoch 9. The descent rate of the loss function gradually slows as the number of epochs increases. Given this observation, training the model past 10 epochs would likely lead to over-fitting.

The testing loss function mirrors the training loss, reaching a local minimum at epoch 8. However, the testing IoU remains largely constant, with a value of 79.9% after 10 epochs of training. This IoU is comparable with values reported by Chen et al. (2017b) which reached peak mIoU values of 81.3% on the Cityscapes test set and 85.7% on the PASCAL VOC 2012 test set with the DeepLabv3 architecture.

The testing IoU value reported here could be improved in two ways: 1) Increasing the number of images in the labeled dataset 2) Reducing image downsampling. Increasing the number of images in the training dataset is feasible, but would require scanning additional persons with limb loss and manual segmentation. As with other forms of medical imaging, limited data quantities and

expert annotations present hurdles for deep learning techniques Wang et al. (2018). On the other hand, reducing the image downsampling could improve the model performance without requiring additional annotations. In Fig. 4 it can be observed that the masks generated by the neural network are partially limited in accuracy due to this downsampling. Still any reduction in downsampling would likely be accompanied by an increased computational cost.

## 4.2 Photogrammetry of Segmented Scans

Table 3: Summary of Cloud to Mesh (C2M) Measurement Count, Mean Absolute Error (MAE) and Root Mean Squared Error (RMSE) of 3D models rendered as part of this study. Models with failed renders from automated workflow marked N/A.

| All Scans | C2M Measurement Count | MAE (mm) | RMSE (mm) |
|---|---|---|---|
| Scan 1 | 149234 | 1.56 | 3.16 |
| Scan 2 | N/A | N/A | N/A |
| Scan 3 | 113706 | 0.06 | 1.26 |
| Scan 4 | 86748 | 0.29 | 2.19 |
| Scan 5 | 197356 | 0.28 | 2.16 |
| Scan 6 | 104333 | 0.00 | 1.49 |
| Scan 7 | N/A | N/A | N/A |
| Scan 8 | N/A | N/A | N/A |
| Scan 9 | 74548 | 0.25 | 4.32 |
| Scan 10 | 110851 | 0.08 | 2.23 |
| Scan 11 | 94102 | 0.05 | 1.58 |
| Scan 12 | 118626 | 0.46 | 5.54 |
| Scan 13 | 78901 | 0.17 | 1.26 |
| Scan 14 | N/A | N/A | N/A |
| Scan 15 | 73525 | 1.15 | 1.84 |
| Scan 16 | 137258 | 0.06 | 3.31 |
| Scan 17 | 134542 | 0.16 | 2.74 |
| Scan 18 | 117213 | 0.01 | 1.76 |
| Scan 19 | 66037 | 1.29 | 1.85 |
| Scan 20 | 110057 | 1.44 | 2.19 |
| Scan 21 | 65703 | 2.02 | 2.74 |
| Scan 22 | 105424 | 0.93 | 2.13 |
| **Average** | **107680 ± 33300** | **0.57 ± 0.63** | **2.43 ± 1.10** |

Table 3 lists the Mean Absolute Error (MAE) and Root Mean Squared Error (RMSE) for the 22 limb scans rendered as part of this study. The average MAE values obtained are excellent (0.57 mm ± 0.63 mm) and comparable with the sub-millimeter MAE accuracy reported by Barbero-García et al. (2020). The RMSE values are higher than those reported by Nightingale et al. (2021; 2020) with an average deviation of 2.43 mm ± 1.10 mm. Scans 1-11 using the bare liner had slightly better MAE and RMSE values (0.32 mm ± 0.51 mm; 2.30 mm ± 1.01 mm, respectively) than scans 12-22 with a colored sock over the bare liner (0.77 mm ± 0.70 mm; 2.54 mm ± 1.21 mm). However the use of the colored sock improved tie point detection and image alignment leading to a successful render rate of 90.9 % for the colored sock vs 72.7% for the bare liner. Scans 2, 7, 8 and 14 did not render properly with the automated workflow. Failure was primarily due to failed photo alignment which led to incomplete and/or distorted models.

Looking at Fig. 5, it can be observed that the high RMSE values are primarily due to slight flexure of the limb between scans and not due to noise introduced by imperfect segmentation. Fig. 5B shows the CloudCompare results comparing Scan 12 to the ground truth. Scan 12 had a low MAE value (0.46 mm) but the highest RMSE value (5.54 mm) of all the scans. It is clear that high RMSE value is due to a change in limb geometry from flexure since the largest variations happen at the knee joint and at the distal end of the limb. This result was obtained even after controlling axis alignment and rotation of the models via the Iterative Closest Point (ICP) algorithm. This pattern is reflected in many other scans, see Fig. 5D and Fig. 5E.

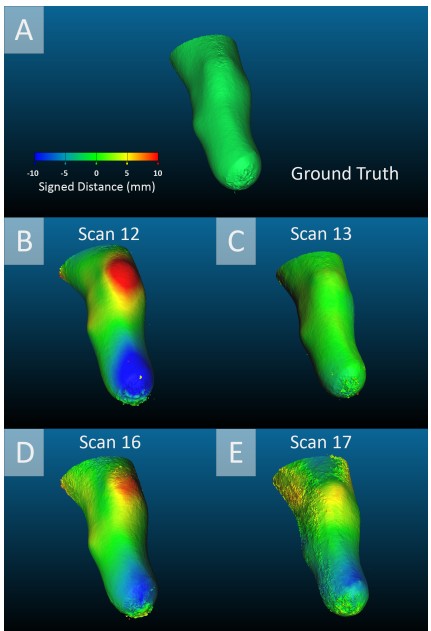

Figure 5: Selected comparison of limbs in CloudCompare. In these 3D scans, it is evident that the large RMSE is due to a change in geometry from limb flexure, and is not due to segmentation or render quality. A) Ground truth model rendered from manually segmented images. B) Scan 12 - Highest RMSE (5.54 mm) evident limb flexure C) Scan 13 - Low RMSE (1.26 mm) no limb flexure D) Scan 16 - Medium RMSE (3.31 mm) evident limb flexure E) Scan 17 - Medium RMSE (2.71 mm) evident surface noise and limb flexure.

Poor surface finish, as evidenced by Fig. 5E, contributes far less to the MAE and RMSE values than flexure of the limb. This is important because poor surface finish can be directly tied to the quality of image segmentation, with better segmentation leading to smoother finishes. Although the accuracy is slightly diminished due to noise, it is not the main cause of error in these scans. Poor surface finish can be accounted for in practice by utilizing Laplacian smoothing functions. Fig. 5C is an example of a limb with low MAE (0.17 mm) and low RMSE (1.26 mm) values. Even without Laplacian smoothing, this model shows remarkable rendering accuracy in comparison with the ground truth.

Since the RMSE errors are systematically impacted by limb flexure, MAE is the more appropriate measure of accuracy in these scans. For the purposes of rendering residual limb models, submillimeter accuracy presents little meaningful advantage to clinicians and practitioners. Unlike the skulls studied by Barbero-García et al. (2017; 2020), amputated limbs undergo constant shape and deformation changes due to effects of donning-doffing, muscle contractions, interstitial fluid movements etc. Suyi Yang et al. (2019); Solav et al. (2019). Traditional methods utilizing Plaster of Paris (PoP) casting as well as clinical gold standards utilizing MRI struggle with shape capture repeatability Safari et al. (2013). Clinicians account for at least a 5-9% volumetric variation in limb size and accommodate for it in socket design Suyi Yang et al. (2019). Given this context, variation in limb shape and size between scans is to be expected.

## 4.3 CLINICAL IMPACT AND TELEMEDICINE DIRECTIONS

The advances in photogrammetric limb scanning resulting from automation of the segmentation step could have potentially significant consequences. Notably, automatic segmentation decreases the amount of human time and effort required to render photogrammetric models. As mentioned previously in the methods, a limb scan captured via the Lim(b)itless smartphone application captures at least 216 images. Manual segmentation of those images would take (at minimum) 1 minute per image leading to a segmentation time of 12,960 seconds. The fine-tuned network trained in this paper would only take 675 seconds to perform this task based on the segmentation rate reported in Table 1. This step is 19.2x faster and requires no human effort, demonstrating a remarkable increase

in performance. Evaluating the end to end process using the rate values in Table 1 reveals that the model rendering workflow is 4.88x faster overall. Although the process bottleneck shifts from segmentation rate to rendering rate, the only step requiring significant human input is the scanning stage which takes less than 1 minute on average, Table 1.

Models from the updated photogrammetric workflow could be used in a variety of contexts, since there was no significant geometrical difference between ground truth models and models rendered from images segmented by the fine-tuned neural network. Most notably, photogrammetric models built from smartphone scans can facilitate a host of low-cost telemedicine solutions. As one example, the limb models could be utilized as the first stage of digital prosthetic socket rectification. This would enable patients to be able to be scanned outside of a clinical environment (eg. at home) and have a prosthesis fabricated digitally and shipped to them. Aside from this, one distinct advantage of the photogrammetric process is it retains surface features which can be reprojected onto the final 3D model. With UV texture mapping, doctors could get a 3D view of limbs and look closely at their outer surfaces (such as for lesions) in virtual reality (VR). Photogrammetric limb scanning could also facilitate long-term and large scale studies of limb shape and size fluctuations over time. Being able to capture data outside of a clinical environment could help clinicians to increase the number of data points and the frequency of measurements beyond what is currently possible.

Several limitations would need to be overcome before using this technology at scale. On a technical level, the fine-tuned segmentation model from this research would need to be evaluated for robustness. Obtaining more scans of different skin colors, different lighting environments, etc. while training the model further could improve the robustness and reduce the risk of bias. It remains to be seen whether this model could be applied to segment different types of amputation or if the model would need to be trained on additional examples. Finally, as noted in Paxton et al. (2022), privacy and regulatory requirements will be the largest concern in implementing a telemedicine system at scale utilizing smartphone photogrammetry. Any healthcare solution would need to comply with the appropriate regulatory guidelines for telemedicine.

## 5 CONCLUSION

This research sought to address limitations in photogrammetric workflows by automating image segmentation via a fine-tuned deep learning model. Some important conclusions are summarized here:

- After 10 epochs of transfer learning, the fine-tuned deep learning model was able to achieve an IoU of 79.9% on the testing training set. The model's loss function indicated that it had reached a local minimum, thus the IoU could not be improved without the risk of overfitting. IoU could be improved by increasing the number of labeled training examples which could also increase model robustness.

- Applying this fine-tuned model to segment a dataset of 22 scans containing 6312 images showed that there was no significant geometric difference compared to scans rendered from manually segmented images. The MAE was found to be on average 0.57 mm ± 0.63 mm while the RMSE was found to be on average 2.43 mm ± 1.10 mm. The error reported was mostly influenced by slight changes in limb shape between scans and not render noise caused by image segmentation.

- Using the fine-tuned model increased the rate of image segmentation by 19.2x as well as sped up the entire photogrammetric workflow by a factor of 4.88x. This performance increase is remarkable since the step requiring the largest human effort was completely automated.

- This fine-tuned deep learning model can help facilitate large scale image processing for telemedicine applications. Using smartphones to acquire 3D scans of amputated limbs, clinicians could monitor patients, conduct large scale research studies, and even build prostheses remotely. This could potentially increase the standard of care for persons with limb loss who do not live near a clinic by matching patients with clinicians seeking to expand their geographic service area.

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
