# OpenReview forum: "Image Segmentation using Transfer Learning with DeepLabv3 to Facilitate Photogrammetric Limb Scanning"
_ICLR.cc/2023/Conference — Submitted to ICLR 2023_

### Official Review · Reviewer_FWEb · 2022-10-23

**Confidence:** 5
**Correctness:** 3
**Technical Novelty And Significance:** 1
**Empirical Novelty And Significance:** 2
**Recommendation:** 3

**Clarity, Quality, Novelty And Reproducibility:**

As mentioned above, while the paper is overall fairly clear there are a number of important details that should be clarified. The quality of the work judged from a technical view point is too low for an ICLR submission. While there is some practical value, the authors only applied standard deep learning tools to their dataset and processed them with another 3D rendering software. There are also no clear empirical insights, e.g. does the model capacity vs image dataset size influence the outcome? are there any augmentation strategies that should be considered? etc.

**Details Of Ethics Concerns:**

There are potential issues about the data split between training and testing, which is not clearly defined to separate images from the same subjects.

**Strength And Weaknesses:**

**Strengths**
- The paper is easy to follow and contains nice visualisation
- The pipeline is evaluated not only for the segmentation quality but also its downstream task: 3D rendering
- There are some interesting qualitative descriptions of the results

**Weaknesses**
- There is little to no technical novelty, all employed methods have been previously presented and are simply applied to a new task
- The description of the dataset and train/validation split is slightly confusing: it is stated that the "fine- tuning dataset consisted of 806 manually labeled images" and further "we randomly select 80% of the images as our training data set and reserve the remaining 20% for validation" does this mean the same subject can be part of training and validation?
- Later on the authors state: "Of the 24 limb scans acquired in the previous step, two scans were manually segmented via Adobe Photoshop and set aside as ground truth references. The remaining 22 scans, totaling 6312 images, were fed into the fine-tuned neural network for segmentation.". Are those different subjects to the ones used for fine-tuning? Furthermore, since 24 scans correspond to 12 subjects with or without additional sock, again the question of whether the 22/2 split is done on a subject-level?
- Overall the low number (2) of held-out ground truth cases makes it somewhat  hard to trust those results, since some cherry-picking might be unavoidable
- All models converge very quickly and no data augmentation seems to be used, this could indicate some overfitting
- an additional practical problem of the pipeline seems to be failure of image alignment that caused scans 2, 7, 8 and 14 to not render properly with the automated workflow. I wonder whether the authors have attempted to use the automatic segmentation information or some other features from the fine-tuned network to improve on this?
- the inference times for automatic segmentation seem awfully slow (0.32 images/s), even when being restricted to Google Colab infrastructure measuring the feed-forward path of a DeepLabV3 with mixed precision (AMP) on a Tesla T4 should result in at least two orders of magnitude faster throughput

**Summary Of The Paper:**

The paper proposes the straightforward application of transfer learning for image segmentation for limb images using DeepLabV3. In addition to the standard IoU metric the authors also rendered 3D models and compared those of the automatic pipeline with manual segmentations. As expected the fine-tuned DeepLabV3 can produce similar quality segmentation and renderings as manual annotations.

**Summary Of The Review:**

I recommend to reject the paper due to limited (or non-existing) technical novelty and low empirical insights.

---

### Official Review · Reviewer_xaFj · 2022-10-24

**Confidence:** 3
**Correctness:** 3
**Technical Novelty And Significance:** 2
**Empirical Novelty And Significance:** 2
**Recommendation:** 3

**Clarity, Quality, Novelty And Reproducibility:**

This paper is well-written and easy to follow. The idea is interesting. Experiments are presented with details.

**Strength And Weaknesses:**

Strength
+ An interesting work that can be very useful for telemedicine or point-of-care diagnosis
+ A workable system that uses a set of existing methods and technologies
+ Good validation

Weaknesses
- Technical novelty is not significant
- Lack of technical comparisons


**Summary Of The Paper:**

This paper proposes a method using deep learning with photogrammetry for scanning amputated limbs. The proposed method uses deep convolutional neural networks, DeepLabv3, and transfer learning. The ResNet-101-based Deep Labv3 model was pre-trained and then fine-tuned. The scan images were acquired by using a smartphone and subsequently segmented on a desktop. Afterward, 3D limb models were rendered from the segmented images using Agisoft Metashape.

**Summary Of The Review:**

The paper presents an interesting idea and good experimental results. However, the method novelty and evaluation needs improvement.

---

### Official Review · Reviewer_bNhp · 2022-10-24

**Confidence:** 5
**Correctness:** 3
**Technical Novelty And Significance:** 1
**Empirical Novelty And Significance:** 2
**Recommendation:** 3

**Clarity, Quality, Novelty And Reproducibility:**

The work is clear, but the novelty is limited in my opinion. It's an interesting confirmation of known techniques, applied to another dataset. The reproducibility of the work may be difficult due to the use of the dataset, which hasn't been made available publicly.

**Strength And Weaknesses:**

The problem statement is clearly explained and the method addresses an important problem of improving time and accuracy of healthcare procedures. A very detailed discussion of clinical impact and directions is presented at the end of the work, what helps with understanding potential applications, advantages and limitations of the method. A few ideas for improvements are also presented, showing that authors have a good understanding of the presented medical problem.

Unfortunately, the novelty of the work is limited. Very similar studies using deep learning segmentation topologies for improvements of photogrammetry have already been conducted. The work is an application of a known method to another computer vision dataset, but experiments focus only on a single model, so it may be difficult to draw some conclusions for future enhancements of such methods.

Also, the reason for selection of the specific DeepLab model hasn't been justified. It would be interesting to verify other models as well.

It's also mentioned that photo alignment led to some failure cases. There are various alignment techniques that could be used to improve this outcome. Have you tried any of them?



**Summary Of The Paper:**

The work proposes to improve the pipeline of amputated limbs scanning by using the deep learning-based image segmentation before rendering them with photogrammetry. In this way, the density of obtained samples can be increased to improve coverage and accuracy of 3D models with more samples. The proposed method is verified in practice in the limb scanning procedure followed by DeepLab model fine-tuning and inference and finally rendering of models from segmented images.  The discussion of achieved results is also well done.

**Summary Of The Review:**

The application is interesting, however the technical contribution is not very significant. Existing methods have been reused for processing of the different dataset, but experiments are limited to some specific configuration only. A more detailed analysis of different segmentation models and preprocessing algorithms to improve limitations such as image alignment would help to improve the quality of the work. In addition, some optimizations of the segmentation model and the rendering procedure could be done to make the entire system suitable for the mobile device.

---

### Decision · Program_Chairs · 2023-01-20

**Decision:**

Reject

**Justification For Why Not Higher Score:**

All reviews agree on rejection of the paper.

**Justification For Why Not Lower Score:**

N/A

**Metareview: Summary, Strengths And Weaknesses:**

The following weaknesses are highlighted:
- All reviewers mention that the paper has limited novelty. The paper applies an existing method to a new dataset.
- The paper is not sufficiently compared to alternative methods.